# Cost-Effectiveness of Immune Checkpoint Inhibitors in Urothelial Carcinoma—A Review

**DOI:** 10.3390/cancers14010073

**Published:** 2021-12-24

**Authors:** Arman S. Walia, Randy F. Sweis, Piyush K. Agarwal, Andrew K. Kader, Parth K. Modi

**Affiliations:** 1Department of Urology, University of California San Diego, La Jolla, CA 92093, USA; kkader@health.ucsd.edu; 2Department of Medicine, University of Chicago, Chicago, IL 60612, USA; rsweis@medicine.bsd.uchicago.edu; 3Department of Surgery, University of Chicago, Chicago, IL 60637, USA; pkagarwal@uchicago.edu (P.K.A.); modi@uchicago.edu (P.K.M.)

**Keywords:** bladder, urothelial, cancer, cost, value, immunotherapy, systemic

## Abstract

**Simple Summary:**

Urothelial carcinoma is a malignancy that originates in the genitourinary tract. It is a heterogeneous disease that can present at different stages, and the treatment options vary in efficacy. Advances in immunotherapy stimulated adoption in urothelial carcinoma, and published trials have shown promising results when compared to conventional therapies. However, oncologic drugs are historically costly, and immunotherapy is no exception. A cost-effectiveness analysis is a standardized method of weighing the clinical benefits of an intervention against the financial burden to obtain a composite proposed value. Multiple investigators have assessed immunotherapy in urothelial carcinoma, but no consensus has been reached. Here, we aim to review the literature of the available cost-effectiveness studies to summarize the results and determine the current value of systemic immunotherapy compared to standard treatment. Positive findings will support continued efforts to adopt immunotherapy, whereas negative findings will identify potential gaps for improvement in cost-effectiveness.

**Abstract:**

Over the last decade, an increasing number of immune checkpoint inhibitors (ICIs) have been assessed for therapeutic efficacy in urothelial carcinoma (UC). The high cost has prompted multiple cost-effectiveness analyses for the various disease stages, with no established consensus. We reviewed the literature to assess the available cost-effectiveness studies and summarize their findings. Studies were filtered for a calculated incremental cost-effectiveness ratio (ICER) to standardize comparison. Over 2600 articles were narrowed to eight primary investigations: one for BCG-refractory non-muscle invasive (NMI), one for neoadjuvant therapy in muscle-invasive (MI), and six for advanced disease. Cost-effectiveness was not achieved for NMI disease. Atezolizumab met the willingness-to-pay (WTP) threshold as neoadjuvant therapy for MI disease compared to chemotherapy, but with multiple limitations on the interpretation. Of the six studies on advanced disease, the results were mixed. This was at least partially attributable to varied methodologies including extrapolated time horizons, inconsistent cost inputs, and different WTP thresholds. Overall, the aggregate results were not compelling enough to establish ICIs as cost-effective compared to conventional chemotherapy. Value may improve with continued investigation into long-term outcomes, refined patient selection, and pricing discounts.

## 1. Introduction

Urothelial cancer (UC) is a commonly diagnosed malignancy, with an estimated 550,000 new cases diagnosed worldwide in 2018 [1]. UC presents with a wide spectrum of disease stages and presentations, ranging from small, non-invasive, low-grade papillary tumors to diffuse metastatic disease. Treatment options vary depending on the grade and extent of tumor invasion. Certain treatments can be effective for oncologic control; however, there is an unmet need for treatments that provide a durable cure. For example, non-muscle invasive (NMI) carcinoma in situ (CIS) of the bladder harbors a 66% risk of progression to muscle-invasive (MI) disease, which intravesical therapy only partially mitigates to 10–20% [2]. Similarly, most patients on first-line treatment for metastatic UC will ultimately progress. National committee guidelines still resort to recommending clinical trial enrollment for this population as even with modern therapies, the median survival remains less than 2 years [3,4].

Immune checkpoint inhibitors (ICIs) emerged from the recognition that tumor cell evasion of the host response can occur by mimicking inhibitory signaling of healthy cells [5]. The three most evaluated checkpoint targets with FDA approval are programmed cell death protein-1 (PD-1), programmed death ligand-1 (PD-L1), and cytotoxic T-lymphocyte-associated protein 4 (CTLA-4). The first approved ICI for the treatment of UC was atezolizumab, which received accelerated approval in May 2016 (ultimately rescinded in 2021). Thereafter, the PD-L1 inhibitor avelumab followed by PD-1 inhibitors nivolumab and pembrolizumab were introduced. The CTLA-4 inhibitor ipilimumab has been evaluated in UC alone and in combination with nivolumab but has not yet received regulatory approval.

Multiple clinical trials have evaluated the use of ICIs for the treatment of various UC disease states, including the following notable examples. KEYNOTE-057 evaluated pembrolizumab for NMI Bacillus Calmette-Guerrin (BCG)-unresponsive CIS in a single-arm phase two study [6]. Forty-one percent had complete response (CR) at three months with a median duration of CR of 16 months. For patients with muscle-invasive disease, the PURE-01 trial evaluated the efficacy of neoadjuvant pembrolizumab [7]. Pathologic CR at the time of radical cystectomy (RC) was 42%, which improved when stratified by a combined positive score (CPS) ≥10% (a ratio of tumor and immune cells expressing PD-L1 to all tumor cells). Atezolizumab and nivolumab have also been explored in the neoadjuvant setting. Several ongoing adjuvant trials with anti-PD-1 therapies are underway, with at least one reporting a recurrence-free survival benefit [8]. KEYNOTE-052 explored pembrolizumab as a first-line treatment for locally advanced or metastatic disease [9]. A 28% response rate was noted with a median treatment duration of 30 months. Overall survival (OS) increased from 11 to 18 months when stratified by PD-L1 expression. KEYNOTE-045 investigated pembrolizumab as a second-line treatment for metastatic disease and recently updated results reaffirmed an improvement in the response rate and OS compared to chemotherapy [10]. The above examples highlight the dynamic role of ICIs in UC as they continue to be investigated.

As with any new intervention, the improvement in outcomes must be balanced with the added cost. Cost-effectiveness analysis (CEA) is a standardized method of reporting and comparing interventions to better inform governing bodies on their added value for appropriate resource allocation. A new therapy with a large clinical benefit and minimal cost is more likely to be adopted than an expensive one with marginal benefit. CEAs are applied throughout healthcare but remain particularly relevant in oncology, where innovation constantly challenges standard therapy and costs are high compared to benign disease. This is supported by increased incorporation into disease management algorithms as demonstrated by the American Society of Clinical Oncology (ASCO) Value Network initiative. Public payors, such as the UK National Health Service, already have an established history of utilizing CEAs to determine medical coverage. The growing constraints on healthcare resources in the United States and abroad will only further the adoption of CEAs into health policy to maximize outcomes and minimize costs.

The primary objective of this review is to assess the current literature on the cost-effectiveness of ICIs in UC to provide a value-based assessment of their contribution to disease management and identify gaps in economic evaluation to guide future investigations.

## 2. Materials and Methods

### 2.1. Search Strategy

A literature search was completed on 1 May 2021, using the PubMed database to identify primary investigations into the outcomes of immunotherapy in bladder cancer and upper tract disease, with a focus on associated costs. Search terms included “bladder” or “upper tract” or “urothelial” or “transitional” and “immunotherapy” or “immune” or “checkpoint inhibitor” with no date limits applied. The search, review, and analysis were performed by one reviewer (A.W.).

### 2.2. Data Selection

There are various methods to assess cost-effectiveness. One example is cost utility (CU). CU includes an assessment of the quality of life in addition to life-years gained. It assigns a utility value (i.e., 0 = death, 1 = perfect health) that when combined with life-years gained creates a composite variable termed quality-adjusted life-year (QALY). The difference in financial investment per QALY is then calculated to determine an incremental cost-effectiveness ratio (ICER), which is one of the more common comparative measures to assess economic value. Studies were filtered to include only those with a reported ICER to select for cost assessments in a standardized and therefore comparable fashion.

Exclusion criteria included duplicates, non-human studies, abstracts only, reviews, commentaries, and non-English language studies. Intravesical BCG studies were frequently encountered as a form of local immunotherapy but were excluded as BCG is not systemic treatment. Remaining studies were reviewed for a discussion of cost or value, and ultimately included if an ICER was reported.

### 2.3. Data Extraction and Analysis

The primary outcome of interest was the incremental cost-effectiveness of systemic immunotherapy compared to the standard of care as conveyed by ICER and the willingness-to-pay (WTP) threshold. The WTP threshold is a pre-determined monetary cutoff for ICER that helps identify which services meet or exceed their accepted valuation. Background variables obtained were author, year of publication, country, type of immunotherapy, and control treatment. Economic variables recorded were currency, analytic method (i.e., Markov, partitioned survival), time horizon, total cost, QALY, and payor perspective. US Dollars ($) were considered the standard, but non-dollar reports were not converted. The input values, such as the average selling price of medication or costs of hospitalization, vary by country, meaning conversion of the ICER into dollars would not provide an accurate comparison across studies conducted in different regions. Instead, meeting the country-specific WTP thresholds was used to gauge value.

Variables of interest, ICER, and WTP were tabulated for each study and categorized by stage of disease: non-muscle invasive (NMI), muscle-invasive (MI), upper tract (UT), and locally advanced/metastatic. Articles on neoadjuvant immunotherapy were categorized as MIBC unless otherwise specified in the inclusion criteria.

### 2.4. Quality and Risk of Bias

A review by Watts and Li on the role of quality in economic evaluations was referenced to guide the selection of an appropriate instrument for risk of bias and quality assessment [11]. The Consensus on Health Economic Criteria (CHEC)-list was chosen for its widespread application and incorporation of assessment of economic modeling [12]. This is a 20-point questionnaire used to critically appraise each study for clear reporting of items, such as the time horizon, appropriate payor perspective, adequate cost evaluation, etc. Each point was scored as 0 or 1, for a maximum of 20.

## 3. Results

The initial search yielded 2687 articles (Figure 1). After applying the inclusion and exclusion criteria, eight primary investigations were included for review.

### 3.1. Quality and Risk of Bias

On the 20-point extended CHEC-list, 6 received a score of 19 and 2 of 18. Four of the six lost a point for not discussing ethics or distributional issues and two reported relevant conflicts of interest. The 2 studies that scored 18 faltered in both categories. Only one of the four conflicts of interest reported a positive result that aligned with direction of the disclosure in question.

### 3.2. Primary Outcome

Of the eight studies included for final analysis, six involved advanced/metastatic disease and two involved localized disease. No studies addressing upper tract disease were identified. General study characteristics for localized disease are outlined in Table 1 and advanced disease in Table 2.

### 3.3. Non-Muscle Invasive

Wymer et al. provided the only study for NMI disease with the primary objective of comparing pembrolizumab to radical cystectomy (RC) and intravesical chemotherapy in BCG-unresponsive CIS tumors [13]. The inspiration was KEYNOTE-057, which prompted FDA approval of pembrolizumab in this population [6]. Data was analyzed via a Markov health state model with the above treatment options in two index populations: (1) patients willing and eligible for RC, and (2) patients unwilling or ineligible for RC. Pembrolizumab was administered at 200 mg q3weeks for up to 24 months. Intravesical gemcitabine-docetaxel (GD) was given as a six-week induction followed by monthly maintenance up to 24 months. Health states were surveillance, recurrence, progression to MI disease, metastasis, treatment toxicity, or death. Costs were calculated using a Medicare payor perspective and Medicare payment schedules utilized for cost inputs. The primary outcome was assessment of the 5-year cost-effectiveness of each treatment option in both index populations. The ICER of pembrolizumab to RC and GD for index 1 was $1,403,008/QALY and $2,011,923/QALY, respectively. Index 2 was $1,073,240/QALY for pembrolizumab to GD. Neither met the pre-established WTP of $100,000. Sensitivity analysis for index 1 demonstrated that pembrolizumab became cost-effective relative to RC after at least a 93% decrease in price. For index 2, a similar reduction of nearly 90% was required to meet the target threshold.

### 3.4. Muscle-Invasive

There was only one study for MI disease by Khaki et al. that evaluated ICIs as neoadjuvant therapy [14]. ICI was compared to chemotherapy (ddMVAC with G-CSF, gemcitabine/cisplatin) using the results and treatment templates provided by the PURE-01 (pembrolizumab), ABACUS (atezolizumab), and NABUCCO (nivolumab/ipilimumab) trials [7,21,22]. A decision tree model was created with three nodes: treatment drug, presence of pathologic CR at the time of RC, and recurrence at two years. The probability of CR was derived from the respective ICI trials. Recurrence-free survival (RFS) and overall survival (OS) values for ICIs were not specifically reported, so 2-year RFS was extrapolated from chemotherapy to each ICI depending on the presence of complete response (CR) at the time of RC. Two-year RFS instead of QALY was selected as the measure of effectiveness in calculating the ICER from the perspective of a third-party payor. Centers for Medicare and Medicaid Services (CMS) average sales price was used for cost inputs. No discount was applied given the shorter 2-year time horizon. ICER of pembrolizumab to ddMVAC was $522,143 and increased to $1,225,058 when compared to gemcitabine/cisplatin (GC). Nivolumab/Ipilimumab ICERs were $1,464,119 and $1,662,327 for ddMVAC and GC, respectively. Atezolizumab demonstrated a similar elevated ICER with GC at $1,629,855. Compared to ddMVAC, atezolizumab showed cost savings at a marginally improved RFS, resulting in a dominant ICER. On sensitivity analyses, the highest variation came from the cost of ICI, probability of two-year RFS, and cost of ddMVAC. Ultimately, only the atezolizumab to ddMVAC comparison met the established WTP threshold of $100,000.

### 3.5. Advanced/Metastatic

Six studies focused on locally advanced or metastatic bladder cancer. Four were on second-line treatment, one on the value of PD-L1 testing, and one on first-line treatment. The main study characteristics are summarized in Table 2. Parmar et al. evaluated atezolizumab vs. taxane chemotherapy as second-line treatment after progression on platinum therapy using a partitioned survival model from a Canadian healthcare system perspective [16]. Outcomes were extrapolated from IMvigor211, which evaluated the two treatments in a similar population [23]. The following three health states were included: progression-free, progression, and death. Cost inputs were obtained from published recommendations of a Canadian expert drug review committee and the Canadian Institute for Health Information’s cost estimator. The time horizon was five years, with cost discounted at an annual rate of 1.5%. ICER for atezolizumab compared to chemotherapy was C$430,652/QALY. In select patients with ≥ 5% PD-L1 expression, ICER improved to C$334,387/QALY. Model variation by the time horizon improved ICER to C$305,408/QALY at 10 years.

Slater et al. investigated ICIs vs. taxane chemotherapy as second-line treatment in patients with progression after platinum therapy [17]. A partitioned survival model was created with three health states: progression-free, progression, and death. KEYNOTE-045 was utilized to guide outcome comparisons between pembrolizumab vs. chemotherapy [10]. Pembrolizumab was administered for up to 2 years on a q3-week regimen based on the trial, and survival estimates were extrapolated to a longer time horizon. Atezolizumab monotherapy was based on IMvigor211 for comparison and administered with a similar frequency for the same period. As no head-to-head trials of the two ICIs existed, the progression-free and progression states were substituted with overall survival (OS) and based on time to day (3,090,180-day windows). Grade ≥ 2 adverse events (AE) and end-of-life care were included due to high associated costs. Drug costs were obtained from wholesale acquisition pricing in the AnalySource database, whereas AE costs were applied from the Agency for Healthcare Research and Quality Health (AHRQ) reports. A 20-year time horizon was used from a third-party payor perspective in the USA with a 3% annual discount applied. Over this period, pembrolizumab extended life expectancy from 0.89 to 2.22 years compared to chemotherapy. This translated to an incremental cost and QALY change of $106,299 and 1.14, respectively, for an ICER of $93,481. Between ICIs, pembrolizumab held a dominant ICER with an incremental cost benefit of −$26,458 for a QALY change of 0.76. The highest impact on ICER in the ICI vs. chemotherapy arm was the cost of pembrolizumab, with the largest variation from extrapolation of OS and the model time horizon. Pembrolizumab’s cost benefit over atezolizumab was driven by the reduced drug cost, with cost variation most pronounced from the OS comparison and treatment duration.

Sarfaty et al. evaluated pembrolizumab vs. taxane chemotherapy as a second-line treatment in multiple countries using a Markov model again based on the KEYNOTE-045 trial [19]. The progression-free, progression, and death health states were used over a 5-year horizon. Cost inputs were based on reported values in each country (no specific sources identified) and included drug, administration, and ≥ grade 3 AE costs. A discount rate of 1.5% was assigned for Canada, and 3% for the rest. A third-party payor perspective was selected for each country. The cost outcomes and WTP threshold were converted to dollars for inter-country comparison. The primary outcome of ICER was calculated as follows: USA $122,557/QALY, UK $91,995/QALY, Canada $90,099/QALY, and Australia $99,966/QALY. None met the assigned country-specific WTP threshold except for the USA, which was set at $150,000. Sensitivity analysis identified extrapolation of OS as the greatest influence on ICER.

Similarly, Srivastava et al. utilized the results of KEYNOTE-045 to investigate pembrolizumab vs. chemotherapy as second-line treatment after platinum. therapy [20]. Vinflunine was included in addition to taxanes. A partitioned survival model was employed with the similar three health states using parametric survival functions to project PFS and OS over a 15-year time horizon. Disease management, drug administration, and ≥grade 3 AE costs were included for analysis. A national Swedish pharmacy was queried for drug acquisition cost. A 3% annual discount rate was applied, and costs assessed from the perspective of the Swedish healthcare system. Incremental cost and QALY for pembrolizumab vs. vinflunine were €69,852 and 1.38 for an ICER of €50,529/QALY. Pembrolizumab vs. taxanes produced an ICER of €81,356/QALY, while pembrolizumab vs. all three chemotherapies yielded an intermediate ICER of €71,924/QALY. At a WTP threshold of €100,000, pembrolizumab demonstrated cost-effectiveness. On sensitivity analysis, the pembrolizumab dose intensity and OS extrapolation had the highest impact on ICER.

Criss et al. took an alternative approach to second-line treatment by focusing on the impact of PD-L1 expression on improving ICER for pembrolizumab vs. chemotherapy (taxanes) [18]. A microsimulation model was created to assess three treatment arms: chemotherapy, pembrolizumab, and pembrolizumab for ≥1% PD-L1 expression otherwise chemotherapy. A discount rate of 3% was applied and costs approached from the perspective of the US healthcare system over a 5-year horizon. The incremental cost and QALY of PD-L1 stratification to chemotherapy was $10,347 to 0.08 for an ICER of $122,933/QALY compared to $197,383/QALY for pembrolizumab in all-comers. This failed to meet the $100,000 WTP threshold but did demonstrate a higher cost-effectiveness in PD-L1-stratified patients. The highest sensitivity of cost-effectiveness was to OS, ICI cost, and utility scores for non-death disease states.

Hale et al. evaluated first-line pembrolizumab vs. gemcitabine/carboplatin (GCa) in cisplatin-ineligible patients whose tumors expressed PD-L1 with a CPS ≥10% [15]. A partitioned survival model with the similar three health states was created using the KEYNOTE-052 trial as a template for PFS and OS estimates [9]. Drug administration, PD-L1 testing, weekly disease management, AEs, and end-of-life care costs were incorporated from public data sets including SEER (Surveillance, Epidemiology, and End Results Program) and the CMS fee schedule. Outcome estimates were extrapolated to a 20-year time horizon using an annual 3% discount rate from the perspective of a US-healthcare third-party payor. Incremental costs and QALY between pembrolizumab and GCa were $158,561 and 2.01 for an ICER of $78,925/QALY, which met the set WTP threshold of $100,000. ICER was most sensitive to time-to-death, discount rate, and cost of disease management for progression two years after treatment initiation.

## 4. Discussion

The relatively recent application of ICIs to urothelial malignancies has continued to show promise as an alternative therapeutic pathway to current management. However, the potential improvement in clinical efficacy must be balanced with the additional cost. The growing importance of cost is well-illustrated by the National Institute for Health and Care Excellence’s (NICE) recent decision to reject coverage of pembrolizumab as it failed to meet their extended end-of-life threshold of £50,000/QALY in the UK [24]. Here, we evaluated eight CEAs categorized by disease stage to highlight the potential for added value at each level.

On quality assessment, each study scored at least 18 of 20 possible points for items considered critical to include on economic evaluations. There is no specific cutoff for “good” or “poor” quality, but the authors of the CHEC list utilized here sought for investigators to attempt to meet each of the 20 unique points when executing cost-based studies [25]. The only points missed on any study were having a relevant disclosure and/or not including a subjective ethics discussion. Disclosures are important as they represent potential conflicts of interest that can certainly introduce bias, but only one of four studies had a relevant disclosure that was aligned with the findings. While an ethics discussion is important, particularly in regard to value-based care, the absence does not adversely affect trial design or outcomes. Ultimately, each study included here met nearly all criteria considered critical to economic evaluation reporting.

The current best practice management of non-muscle invasive bladder cancer varies by risk category. The higher risk categories introduce a more complex clinical dilemma as higher risk of progression and ultimately metastasis must be weighed against the morbidity of cystectomy. KEYNOTE-057 attempted to complement intravesical therapy by evaluating pembrolizumab in BCG-unresponsive NMI UC. The data set the stage for the analysis by Wymer et al., which aptly included the appropriate disease states in their model and separated RC-eligible and ineligible patients into two independently analyzed cohorts. Pembrolizumab was, however, unable to meet the ICER threshold by a wide margin. It is difficult to envision pembrolizumab becoming cost-effective here in the near future as assumptions in the model already favored the ICI (i.e., no progression to metastasis in any patient) such that there would need to be a drastic improvement in outcomes data for a significant impact. Similarly, a > 90% price reduction was needed to meet the threshold. This is a larger discount than is typically seen even with conversion from patented to generic drug pricing. The main limitation of the study was the extrapolation of two-year outcomes to a five-year horizon, a common finding amongst the CEAs reviewed. Alternative approaches to reach cost-effectiveness may be investigating high-risk papillary disease instead of CIS as it may be more sensitive to treatment as well as trialing less expensive ICIs. Ultimately, radical cystectomy established itself as a higher value approach from a payor perspective and despite the morbidity will likely remain so until lower-cost alternatives are identified.

MI UC is encountered on presentation or after progression of NMI disease. The current recommendation is for RC, with consideration of trimodal therapy (TMT) for patients ineligible for surgery. The addition of neoadjuvant chemotherapy has repeatedly shown improvement in downstaging at RC as well as overall survival in this population [26]. Grossman et al. demonstrated a five-year OS of 57% for a neoadjuvant MVAC cohort compared to 43% for RC only, with a higher pathologic CR at the time of RC of 38% vs. 15% [26]. Dash et al. retrospectively compared GC to MVAC and found a similar pathologic CR rate of 26% and 28%, respectively [27]. In both studies, lower pathologic stage at the time of cystectomy was associated with increased overall survival, suggesting similar efficacy of the two agents but with potential lower toxicity in the GC group. The pursuit of an effective neoadjuvant agent with limited secondary toxicity paved the way for ICIs.

In the single-arm phase II PURE-01 trial, patients with MIBC received three cycles of pembrolizumab followed by RC [7]. Overall, 42% achieved pathologic CR at the time of RC with a 2% grade 3 or 4 AE rate. When stratified by tumor PD-L1 expression, 54% of patients with CPS ≥10% demonstrated CR vs. 13% in CPS <10%. The ABACUS trial was a similar phase II single-arm investigation into neoadjuvant atezolizumab [21]. The pathologic CR rate at the time of RC was 31%, with 12% experiencing treatment-related grade 3 or 4 toxicity. Although there was a higher percentage of CR noted in the PD-L1-positive cohort (37%), this was not statistically significant. Van Dijk and colleagues evaluated a combination of nivolumab and ipilimumab in stage III UCC patients and demonstrated a 46% pathologic CR, with 55% of participants experiencing a grade 3 or 4 adverse event [22]. The higher observed AE rate may have been partly due to the more advanced disease in this cohort.

Khaki et al. provided the only CEA for neoadjuvant therapy and was based on the above three clinical trials. The authors were thorough in comparing the three ICIs to both ddMVAC and GC, but only atezolizumab met the WTP threshold compared to ddMVAC due to a marginally higher 2-year RFS. One could argue that the comparison to GC is more clinically relevant as it is an NCCN guideline-supported alternative to MVAC at a substantially reduced cost. However, the mild survival benefits calculated were not enough to outweigh the costs (>$1,000,000 ICER for each compared to GC). Incorporating treatment-related AEs may have partially offset the cost difference in favor of ICIs, but these were not included in the analysis. Atezolizumab’s value is further called into question considering its recent withdrawal as a second-line therapy in advanced UC after publication of follow-up phase III trial data. Given that the ABACUS trial is a phase II study, a similar reversal could occur when future investigations are completed. Lastly, the authors deviated from the accepted standard calculation of ICER by replacing QALY with 2-year RFS. This is further limited in that the RFS rates were extrapolated from pathologic CR. To the authors’ credit, they attempted to contact the clinical trial investigators for appropriate outcomes data. However, 2-year survival data is of particularly limited benefit when evaluating the effect of neoadjuvant therapy, where a 5-year horizon or longer would be more relevant. Overall, the findings show some promise for the future role of neoadjuvant ICIs but also highlight the dependence of CEAs on available outcomes data.

Current accepted guidelines for advanced or metastatic UCC are first-line GC or MVAC, and carboplatin for cisplatin-ineligible patients. Second-line treatment is less concrete and varies from vinflunine, taxanes, to clinical trial enrollment. The lack of robust data in favor of a particular treatment has made it comparatively ripe with studies into the role of ICIs. In KEYNOTE-045, the pembrolizumab cohort had a statistically significant higher OS of 10.3 months from 7.4 months and the difference was maintained in tumors expressing PD-L1 [10]. CheckMate 032 studied nivolumab with and without ipilimumab in a similar population and demonstrated an objective response rate between 26% and 38% with combination therapy [28]. Although IMVigor210 was encouraging for atezolizumab, the most recent update in IMVigor211 failed to meet the primary endpoint and FDA approval has since been rescinded [23].

Second-line treatment was analyzed in four studies, which were split in their ability to meet WTP thresholds. Parmar et al. extrapolated from IMVigor211 and failed to find atezolizumab cost-effective. The lower rate of AEs and longer duration of response for atezolizumab were not enough to make up for the small difference in overall survival and large cost differential. This study was inherently limited as the trial it was based upon did not demonstrate an OS benefit. Slater et al. and Srivastava et al. deemed pembrolizumab cost-effective compared to conventional chemotherapy using KEYNOTE-045 results extrapolated to 20-year and 15-year horizons, respectively. The potentially higher durability of ICI response may indeed result in cost-savings, but without robust long-term survival data, the inferred outcomes can lead to overestimation of benefit. Parmar et al. demonstrate this effect as cost-effectiveness improved when increasing the 2-year trial follow-up to an extrapolated 10-year period. This is further supported by a retrospective review of various CEAs that demonstrated more favorable ICERs were calculated when longer time horizons were used (>5 years) [29]. The pattern held true in the individual studies evaluating both short- and long-term horizons. One must be cautious accepting the results of extrapolated time horizons because a small survival benefit becomes artificially magnified without supporting real-world data. This is particularly relevant in the second-line treatment population where the life expectancy is significantly lower than other UC disease states.

Sarfaty et al. performed a multi-national CEA with an ICER per country for pembrolizumab vs. chemotherapy. The only WTP threshold met was for the USA ($150,000) at an ICER of $122,557. The rationale for setting WTP at $150,000 was not explicit, and deviates from the literature standard of $100,000. Additionally, the authors only included direct medical costs in the analysis without the ongoing disease management or end-of-life care Srivastava et al. incorporated into their analysis. This may have artificially decreased the total costs calculated and indirectly lowered the ICER.

Hale et al. provided the only CEA for first-line treatment and met the WTP threshold when comparing pembrolizumab to GCa in cisplatin-ineligible patients with CPS ≥10%. The strengths of this study included their comprehensive cost inputs (i.e., routine disease management, end-of-life care) and use of updated results from the KEYNOTE-052 trial. Although a longer time horizon is more relevant for first-line treatment, the results should be cautiously reviewed given the 20-year time horizon extrapolated from a median of 29 months of follow up.

Criss et al. modeled PD-L1 testing as a selective treatment strategy and although the WTP threshold was not met, the results trended toward a cost-effective approach. Parmar et al. similarly identified a pattern of increasing cost-effectiveness with increasing selectivity of treatment. PD-L1 stratification has been highlighted in various trials, but the benefit remains unclear. Davis and Patel reviewed the literature on PD-L1 expression as a predictive biomarker for ICIs in a range of tumors [30]. They reported that only 28% of studies found expression to be predictive, and 53% did not. The value of PD-L1 will likely gain further clarity as more studies are published. If PD-L1 expression does not end up becoming a reliable biomarker for effective treatment selection, it is clear that attempts should continue at identifying such a test.

Comparisons between the CEAs were limited by the heterogeneity in methodology. Studies included were preliminarily limited to those with ICERs to attempt standardization. Yet, variations persisted that made it difficult to perform head-to-head comparisons, such as the modeling technique, annual discount rate, and inclusion of additional costs, such as routine disease management or end-of-life care. Some studies extrapolated both treatment arms from the same trial, whereas others performed indirect treatment comparisons requiring an additional set of assumptions. The definition of the WTP threshold was also variable. This is a subjective cut-off that assigns a monetary value to a social good (i.e., $100,000 per QALY for pembrolizumab). In a review by Nanavaty et al., selection of a WTP value was influenced by various factors [31]. The severity of disease mattered as diabetes treatments tended to have a more stringent limit then non-small cell lung cancer. Similarly, patients were afforded more leniency for end-of-life care. Certain countries held lower thresholds than others for the same intervention. Taken together, there is a subjective foundation to this mode of objective assessment. The subjectivity, while posing a limitation to quantitative analysis, appropriately reflects varying views on cost-effectiveness and value. These issues remain complex and may vary from one society to another, as clearly reflected by differences in health policy among countries.

## 5. Conclusions

UC is a heterogeneous disease with a need for more durable treatments. The introduction of ICIs into UC has encouraged a wave of promising outcomes-based studies, which need to be balanced with the higher associated costs. This has secondarily led to CEAs that attempt to gauge the value provided by these new immunotherapies. Though some analyses fell short of the set WTP threshold, the durable response of ICIs and favorable results in certain cohorts demonstrated a trend toward cost-effectiveness when longer time horizons and selective treatment algorithms were applied. There were considerable differences between studies, including the trial data used, the methodology employed, and the definition of value. This limited direct comparison among CEAs. The summarized data shows ICIs are not currently cost-effective in the treatment of UC, regardless of stage. However, improved patient selection, especially with the identification of a predictive biomarker, long-term outcome data, and increased competition leading to more favorable drug pricing, may improve value in the future.

## Figures and Tables

**Figure 1 cancers-14-00073-f001:**
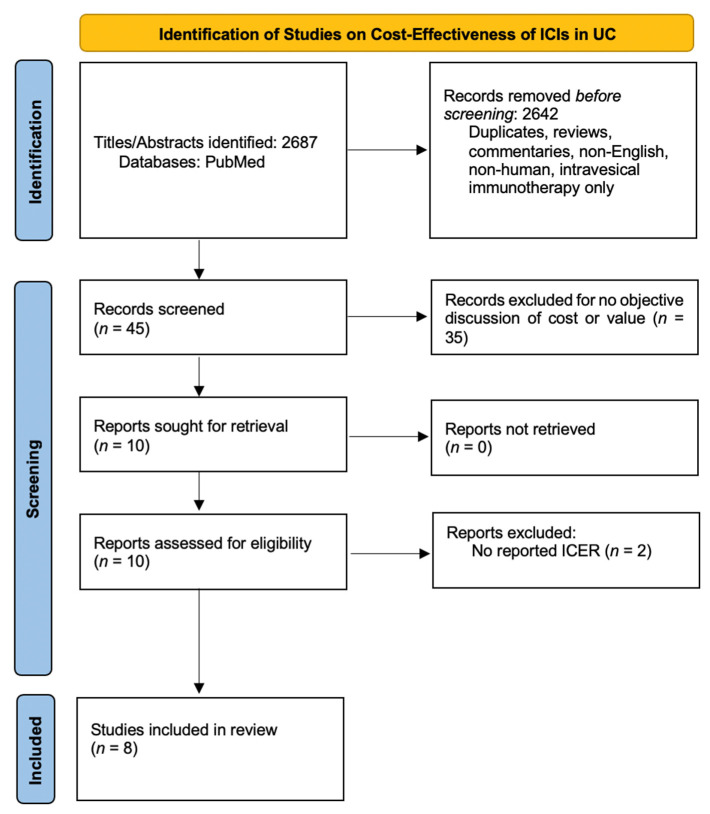
Selection and review of relevant CEAs for ICI in UC.

**Table 1 cancers-14-00073-t001:** CEAs for non-metastatic UC.

Stage	Reference	Cohort	Method	Total Cost	QALY	ICER (Per QALY)	WTP Threshold (Per QALY)	Conclusion
NMIBC	Wymer et al., 2020 [13], USA	Pembrolizumab (PB) vs. intravesical gemcitabine/docetaxel (GD) vs. upfront radical cystectomy (RC) for BCG-refractory NMIBC	Markov Model, 5-year horizon, Medicare payor perspective, base year 2019	Index 1 PB: $191,297 GD: $43,488 RC: $39,367 Index 2 PB: $189,902 GD: $38,192	Index 1 PB: 4.39 GD: 4.32 RC: 4.29 Index 2 PB: 4.35 GD: 4.21	Index 1 PB/RC: $1,403,008 PB/GD: $2,011,923 Index 2 PB/GD: $1,073,240	$100,000	PB is not cost-effective compared to RC or intravesical GD for BCG-refractory NMIBC
MIBC	Khaki et al., 2021 [14], USA *	Neoadjuvant PB vs. ddMVAC. Secondary gemcitabine/cisplatin (GC), atezolizumab (AZ) and nivolumab/ipilimumab (NI) assessment	Decision analysis simulation, 2-year horizon based on RFS given no OS data, third-party payor perspective, base year 2020	PB: $30,556 ddMVAC: $22,515 GC: $910 AZ:$18,838 NI: $74,052	PB: 0.591 ddMVAC: 0.576 GC: 0.567 AZ: 0.578 NI: 0.611	PB/ddMVAC: $522,143 AZ/ddMVAC: Dominates (−$1838/500) NI/ddMVAC: $1,464,119 PB/GC: $1,225,058 AZ/GC: $1,629,855 NI/GC: $1,662,327	$100,000	PB is not cost-effective compared to ddMVAC or GC as a neoadjuvant treatment based on 2-year RFS AZ may be cost-effective compared to ddMVAC, not GC

NMIBC: non-muscle invasive bladder cancer, MIBC: muscle-invasive bladder cancer. * Khaki et al.: 2-year RFS is substituted for calculated QALY, and ICER is incremental cost per 2-year survival rate.

**Table 2 cancers-14-00073-t002:** CEAs in advanced UC.

Treatment	Reference	Cohort	Method	Total Cost	QALY	ICER (Per QALY)	WTP Threshold (Per QALY)	Conclusion
First-line	Hale et al., 2020 [15], USA	First-line PB vs. gemcitabine plus carboplatin (GCa) in cisplatin-ineligible patients with CPS ≥ 10	Partitioned survival model, 20-year time horizon, third-party payor perspective, AEs included, base year 2018	PB: $225,334 GCa: $66,773	PB: 2.91 GCa: 0.90	PB/GCa: $78,925	$150,000	PB may be cost-effective relative to GCa for 1L treatment for PD-L1 positive, cisplatin-ineligible patients with metastatic UC
Second-line	Parmar et al., 2020 [16], Canada	AZ vs. chemotherapy (taxanes) as second line treatment after progression on cisplatin	Partitioned survival model, 5-year horizon, Canadian healthcare system perspective, includes AEs and end-of-life care, base year 2018	AZ: C$90,290 Chemo: C $8,466	AZ: 0.75 Chemo: 0.76	AZ/Chemo: C $430,652 AZ/Chemo: C$334,387 (for subgroup PDL1 expression > 5%) AZ/Chemo: C $305,408 (for extended 10-year horizon)	C$100,000	AZ is not considered cost-effective relative to taxanes as 2L treatment for advanced UC ICER improves with PD-L1 stratification and longer time horizon
	Slater et al., 2020 [17], USA	PB vs. chemotherapy (taxanes) after progression on first-line therapy Secondary analysis comparing PB and AZ	Partitioned survival model, 20-year horizon, third-party payor perspective, base year 2018	PB: $140,556 Chemo: $34,257 PB: $152,753 AZ: $179,211	PB: 1.79 Chemo: 0.66 PB: 1.79 AZ: 1.03	PB/Chemo: $93,481 PB/AZ: Dominates (−$34,813)	$100,000	PB may be cost-effective compared to taxanes for advanced UC Indirect comparison suggests PB more cost-effective than AZ
	Criss et al., 2018 [18], USA	Cost-effectiveness of PD-L1 testing in second-line treatment. Compared chemotherapy (taxanes), PB, and PB for PD-L1 > 1% with chemotherapy for PD-L1 < 1%	Microsimulation model, 5-year horizon, US healthcare system perspective, base year 2017	PD-L1 > 1%: $27,579 Chemo: $17,732 PB: $40,573	PD-L1 > 1%: 0.51 Chemo: 0.43 PB: 0.58	PD-L1 > 1%/chemo: $122,933 PB/PD-L1 >1%: $197,383	$100,000	The cost-effectiveness of PB improves when PD-L1 status is included, but still fails to meet the WTP threshold
	Sarfaty et al., 2018 [19], multi-national *	PB vs. chemotherapy (taxanes) for second-line treatment	Markov model, 5-year horizon, third-party payor perspective, included AEs, base year 2017	* USA: $44,325 UK: $33,271 Canada: $33,869 Australia: $36,154	* USA: 0.36 UK: 0.36 Canada: 0.37 Australia: 0.36	USA: $122,557 UK: $91,995 Canada: $90,099 Australia: $99,966	USA: $150,000 UK: $65,000 Canada: $80,000 Australia: $60,000	PB may be cost-effective compared to chemotherapy at the WTP threshold set for the USA
	Srivastava et al., 2018 [20], Sweden	PB vs. chemotherapy (taxanes, vinflunine) for second-line treatment	Partitioned survival model, 15-year horizon, included AEs, base year 2018	PB: €98,354 Vinflunine: €28,501 PB: €98,348 Taxane: €25,182 PB: €98,208 Chemo: €25,054	PB: 1.99 Vinflunine: 0.61 PB: 1.99 Taxane: 1.09 PB:1.99 Chemo: 0.97	PB/Vinflunine: €50,529 PB/Taxane: €81,356 PB/Chemo: €71,924	€100,000	PB may be cost-effective relative to vinflunine and taxanes

* Sarfaty et al. only reported incremental costs and QALY for each country, not the total. Additionally, the primary authors converted each currency to US dollars.

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
