# Peer review of "Cost-Effectiveness of Immune Checkpoint Inhibitors in Urothelial Carcinoma—A Review"

_cancers, 2021, doi:10.3390/cancers14010073_

Round 1
Reviewer 1 Report
There are still a few lingering concerns that I have.
- (line 104) It would be good to explain how a "comprehensive" literature search can be done using only one academic database.
- (line 108) In the revised manuscript, "cost" is included in the search even though the authors said in the response letter that "We decided to repeat the search more broadly without cost to ensure we did not miss titles that utilized other words such as “value”", so which is correct?
- (Figure 1) Figure 1 looks distorted in the PDF, is this a formatting error?
- (Table 1) Since the authors have decided not to standardize the base year of the currency, can the base year be included for better interpretation please?
Reviewer 2 Report
All the comments have been addressed.
Thank you.
Round 2
Reviewer 1 Report
The authors have adequately addressed the concerns.
This manuscript is a resubmission of an earlier submission. The following is a list of the peer review reports and author responses from that submission.
Round 1
Reviewer 1 Report
The authors conducted a systematic literature review of cost-effectiveness studies determining the value of immune checkpoint inhibitors in urothelial carcinoma according to PRISMA guidelines. The research question is clinically relevant, and the authors thoroughly address the studies and limitations. I have some minor questions and suggestions.
- In Figure 1 flow diagram, the last flow does not add up: after n=10 and excluding 1 study for no reported ICER, the final number becomes n=8. Please check the number of excluded studies.
- In Table 1, please define the stage categories – NMIBC and MIBC – in the footnote.
- In Table 2 and the main texts in the Results section, is the currency used for the Parmar et al. (Canadian health system) study in Canadian dollar?
Thank you.
Reviewer 2 Report
This article reads well but there are many fundamental issues that need addressing.
Based on the PRISMA 2020, the following may need further clarification:
- Why was only one academic database searched?
- What's the eligibility criteria? Any limit on language?
- Why wasn't any words related to "cost" a search term?
- Line 113: The statement isn't true, utility values can be negative
- Protocol of this systematic review has not been registered before this systematic review was completed.
Other major concerns include:
- The title suggest cost-effectiveness analysis studies being searched but authors have only searched for studies that conducted cost-utility analysis. Why not include those expressed in natural units like incremental cost per mortality averted which would be as important as ICER?
- The time horizon of each study differs (extended time horizon tends to yield more favourable ICERs[1]) with different perspective studied (hence different types of cost collected) and different base year for the currency plus WTP threshold differs for different countries. Therefore, is it correct to compare the ICERs without any standardization?
- What is the gap in economic evaluation? Is it correct to suggest gaps when this study has excluded cost-consequences analysis, cost-benefit analysis, cost-minimisation analysis and cost-effectiveness analysis?
- CHEC was used to assess article's quality - was there no within-trial cost-utility analysis (CUA) included or was the exclusion criteria to exclude within-trial CUA? Wouldn't the exclusion of outcome expressed in natural units exclude within-trial CUA and reduce the articles relevant to the research question?
- What if the CHEC item was partially fulfilled - would it have been scored as 0 or 0.5?
- What is the base year of the cost?
- Line 133: If non-dollar reports were not converted, why is the cost in Table 2 for UK in dollar and not UK pounds?
- Table 2: Can the correct currency be presented - is it CAD$80,000 and AUD$60,000 for Canada and Australia respectively or are both in USD? Can the currency code be used to differentiate countries that uses dollar (e.g. Canadian dollar, Australia dollar) please?
REFERENCES
1. Kim DD, Wilkinson CL, Pope EF, Chambers JD, Cohen JT, Neumann PJ. The influence of time horizon on results of cost-effectiveness analyses. Expert review of pharmacoeconomics & outcomes research. 2017;17(6):615-23.